

# A retrospective study of consistency between immunohistochemistry and polymerase chain reaction of microsatellite instability in endometrial cancer

Cheng Wang[1,2,*], Wei Kuang[1,2,*], Jing Zeng[3], Yang Ren[4], Qianqi Liu[1,2], Huanxin Sun[1,2], Min Feng[1,2] and Dongni Liang[1,2]

[1] Department of Pathology, West China Second University Hospital of Sichuan University, Chengdu, Sichuan, China
[2] Key Laboratory of Birth Defects and Related Diseases of Women and Children (Sichuan University), Chengdu, Sichuan, China
[3] Department of Obstetrics and Gynecology, West China Second University Hospital of Sichuan University, Chengdu, Sichuan, China
[4] West China Hospital of Sichuan University, Chengdu, Sichuan, China
[*] These authors contributed equally to this work.

Corresponding author
Dongni Liang, dongni1108@163.com

## ABSTRACT

**Objectives**. Identification of endometrial cancers (EC) with mismatch repair deficiency (dMMR) or microsatellite instability-high (MSI-H) is essential for Lynch syndrome screening and treatment stratification. We aimed to assess the utility of immunohistochemistry (IHC) staining for MMR protein expression and polymerase chain reaction (PCR)-based MSI assays in EC and the correlation between MMR/MSI status and various clinicopathological parameters.

**Methods**. We reviewed the clinical and pathological information of 333 patients with EC. MMR protein expression was assessed as retained or lost to determine MMR status by IHC staining, and MSI status was identified by PCR capillary electrophoresis (PCR-CE) testing with a National Cancer Institute (NCI) panel. The correlation of MMR/MSI status with clinicopathological features was determined by statistical analysis. Discrepant results were further analyzed using an alternative PCR-CE MSI (Promega panel) method, MLH1 promoter methylation assays, and next-generation sequencing (NGS).

**Results**. Among the EC patients, the overall percentage of dMMR was 25.2%, and the overall percentage of MSI-H was 24%. Among the dMMR patients, 50 (59.5%) showed loss of MLH1 and PMS2 expression, 19 (22.6%) loss of MSH2 and MSH6 expression, and seven (8.3%) and eight (9.5%) loss of PMS2 and MSH6 expression, respectively. The dMMR subgroup was significantly younger than the pMMR subgroup, especially for <60-years-old patients ($p = 0.038$). In addition, we identified a strong correlation between MMR/MSI status and high-grade endometrioid or nonendometrioid components ($p = 0.004$ or $p = 0.003$). IHC staining and PCR-CE assay results showed a high level of overall concordance (98.8%, Cohen's $\kappa = 0.98$). Four patients were found to have dMRR/MSS in both examinations. We reanalyzed them with additional methods. One case showed MLH1 promotor methylation, and the other three cases harbored

MSH6 germline pathogenic variations. One of the cases with MSH6 deficiency was reanalyzed as MSI-H by alternative PCR-CE assay or NGS testing.

**Conclusions**. This study indicates that the combined use of MMR-IHC and PCR-CE MSI analyses may effectively avoid misdiagnoses of EC patients with dMMR/MSI-H. However, use of PCR-CE alone to evaluate MMR/MSI status may lead to missed diagnosis, especially for EC patients with MSH6 deficiency and presenting MSS.

## INTRODUCTION

The function of the DNA mismatch repair (MMR) system is to identify and correct errors in DNA replication and ensure fidelity of the replication process (*Li, Pearlman & Hsieh, 2016*; *Ijsselsteijn, Jansen & deWind, 2020*). MMR deficiency (dMMR) refers to loss of function of MLH1, PMS2, MSH2 and MSH6 proteins, resulting in the loss of function of the MMR system, which plays a crucial role in maintaining genomic stability. Microsatellites refer to short tandem repeats distributed throughout the genome. Microsatellite instability (MSI) is a change in microsatellite length due to insertions or deletions in microsatellite repeat units during DNA replication and the inability of the MMR system to correct these errors. MSI is mainly caused by lack of MMR protein expression; therefore, detecting protein deletions can indirectly reflect MSI status (*Bonadona et al., 2011*; *Lee et al., 2022*; *Goverde et al., 2018*).

In 1993, *Ionov et al. (1993)* were the first to identify the presence of MSI as a frequent molecular phenomenon in colorectal cancer (CRC). Subsequently, clinical studies have shown MSI-high (MSI-H) in approximately 2% to 4% of diagnosed cancers, and solid tumors with a high incidence of MSI include endometrial cancer (EC) (17%–33%), gastric cancer (9%–22%), and CRC (6%–13%) (*Goldstein et al., 2014*; *Marabelle et al., 2020*). MSI-H/d-MMR is also clinically meaningful for various aspects of Lynch syndrome (LS) screening, molecular classification, prognosis, adjuvant therapy and immunotherapy (*Mateo et al., 2015*; *Latham et al., 2019*). The American Cancer Genome Atlas (TCGA) classifies EC into four molecular subgroups: polymerase-epsilon (POLE) ultramutated, MSI hypermutated, copy-number low and copy-number high (*Levine & Cancer Genome Atlas Research Network, 2013*). MSI is an essential subtype in molecular classification of EC, with a prognosis between the POLE ultramutated and low-copy types (*Kommoss et al., 2018*). Therefore, molecular classification can facilitate accurate clinical outcome comparisons between different subgroups of patients, further impacting treatment considerations. According to the National Comprehensive Cancer Network (NCCN) guidelines, detection of MMR/MSI status may be recommended for all patients diagnosed with CRC and EC as screening for LS and to identify patients with metastatic disease who might benefit from immune checkpoint inhibitor therapy (*Schrock et al., 2019*; *Zhao et al., 2022*).
The main methods for MMR/MSI include immunohistochemistry (IHC) for MMR proteins, fluorescent multiplex polymerase chain reaction (PCR)-capillary electrophoresis (CE) or next-generation sequencing (NGS) for MSI assays. The IHC technique is usually recommended as a screening strategy in many clinical laboratories because of its low cost and time consumption (*McConechy et al., 2015*). In clinical practice, it has been found that IHC results interpretation is subjective, that IHC results are dependent on the quality of the specimen analyzed, and that some cases of abnormalities caused by other MMR proteins may be missed when using IHC. PCR-CE is the gold standard for determining MSI status and provides an objective assessment of functional dMMR activity, but it cannot directly determine a protein that appears to be functionally abnormal, and it is limited by nucleotide repeat markers and the tumor content (>30%) (*Toh et al., 2021*). In 2018, the European Society for Medical Oncology (ESMO) reported a summary of 16 studies that included 3494 cases, showing that IHC methods have up to 92.4% sensitivity and 99.6% specificity compared to PCR-CE detection (*Vangala et al., 2018*). Thus, dMMR corresponds to the MSI-H phenotype, and MMR proficient (pMMR) corresponds to MSI-low (MSI-L)/microsatellite stable (MSS). However, previous studies have found differences in the concordance between PCR-CE and IHC staining for various solid tumors, such as CRC and EC (*Bartley et al., 2012*; *Wang et al., 2017*). Recently, NGS has emerged and is gradually being used in MSI testing, especially for tumor patients in whom other molecular markers need to be detected simultaneously. However, due to the different NGS platforms, MSI algorithms and positive interpretation criteria have not yet been standardized (*Middha et al., 2017*). Despite the high accuracy of these approaches, the inconsistency between them may lead to misdiagnosis. Primary resistance occurs in 10% to 40% of patients with MSI-H/dMMR metastatic CRC who receive immunotherapy. Some of these cases may be due to misdiagnosis of MSI or MMR, accounting for approximately 10% (*Cohen et al., 2019*). Overall, specific guidance on the preferred approach is lacking.

The present study analyzed clinicopathological characteristics and investigated the concordance between IHC staining and PCR-CE testing of 333 EC patients. We further comprehensively analyzed the reasons for discordant cases of MMR/MSI status and determined the usefulness of the combined assay for assessing MMR/MSI status and the potential risk of misdiagnosis that may exist with PCR-CE testing alone.

## MATERIALS AND METHODS

### Patients and samples

This study retrospectively enrolled 333 patients with endometrial carcinoma who received genomic profiling with MMR protein IHC staining and microsatellite instability-based testing by PCR amplification of specific microsatellite repeats from October 2021 to February 2023 in the West China Second University Hospital of Sichuan University. The patient inclusion criteria are shown in a consort flow diagram in Fig. 1. Primary clinical and pathological data were collected from the patient's medical records and pathological reports, including age, sex, family history (CRC, EC and other cancers), International Federation of Gynecology and Obstetrics (FIGO) stage (2009), histology, grade (nonendometrioid

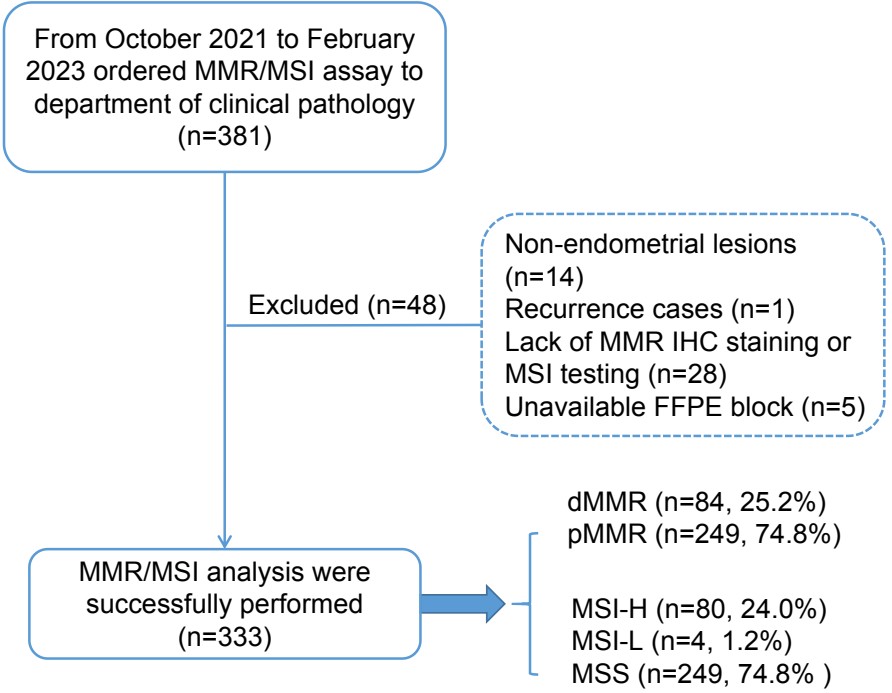

**Figure 1 Consort flow diagram for study inclusion.** MMR, mismatch repair; MSI, microsatellite instability; dMMR, mismatch repair deficient; pMMR, mismatch repair proficient; MSI-H, microsatellite instability-high; MSI-L, microsatellite instability-low; MSS, microsatellite stable.

components were classified as grade 3), lymphovascular invasion (LVSI) and myometrial invasion.

Patients with MSI-H/pMMR or MSS/dMMR were further subjected to additional MSI-PCR (Promega panel) assays, NGS testing, and MLH1 promoter methylation assays to evaluate the cause of discordant cases. The study was approved by the Ethics Committee of West China Second University Hospital of Sichuan University Institutional (No. 2023037). Written informed consent was obtained from all the participants prior for the publication of this study.

## MMR protein IHC staining

IHC staining for MLH1 (1:100; clone ES05; Fuzhou Maxin, China), PMS2 (1:100; clone EP51; Fuzhou Maxin, China), MSH2 (1:1000; clone MX061; Fuzhou Maxin, China), and MSH6 (1:1400; clone MX056; Fuzhou Maxin, China) was performed on 4 $\mu$m-thick tissue sections with Bond-III (Leica Biosystems, Germany) according to the manufacturer's instructions. The result of MMR protein staining was interpreted as retained or lost by two experienced pathologists. The positive signal was located in the nucleus, and the staining intensity of tumor cells was compared with that of internal control cells, including lymphocytes, mesothelial cells, and glandular cells. Tumor tissues with surrounding normal tissues showed nuclear staining, and lack of background nonspecific staining was interpreted as retained staining. Loss of staining was defined as tumor cell nuclei without

staining and normal nuclei of surrounding tissues with staining. Tumors with deficient expression of one or more MMR proteins were defined as dMMR, whereas tumors with positive staining of all four proteins were defined as pMMR. This study did not include cases in which immunohistochemical staining was challenging to interpret.

## DNA extraction

Serial sections were taken from formalin-fixed paraffin-embedded (FFPE) tissue blocks for DNA isolation. All slices of FFPE tumor tissue specimens were used for testing after confirmation and marking by two experienced pathologists using hematoxylin and eosin (H&E)-stained sections to ensure adequate tumor cells (>30%) and no apparent necrotic, mucinous, or inflammatory lesions in the tissue. The tumor areas and paired normal tissues were scraped on slides without coverslips. After 2 h of digestion with proteinase K at 56 °C, followed by 1 h at 95 °C, UPure FFPE Tissue DNA Kit (Biokeyston, Chengdu, China) was used for DNA extraction.

## MSI testing using PCR-CE for fragment analysis

MSI status was determined by PCR-CE analysis of an NCI panel kit (Tongshu BioTech, Shanghai, China) using an ABI 3500dx automated Genetic Analyzer with the GeneMapper® ID-X v.1.6 software (Applied Biosystems, Foster City, CA, USA). This panel, known as the Bethesda panel (NCI panel), contains two mononucleotide loci (BAT25 and BAT26), three dinucleotide loci (D2S123, D5S346, and D17S250), and one pentanucleotide repeat marker (Penta C) as the internal control. Two experienced pathologists analyzed the results. Cases were defined as MSI-H, which presented more than or equal to two instability loci; MSI-L indicated only one locus, and MSS showed no unstable loci.

## Discordant cases evaluated with the Promega panel and NGS test and an additional MLH1 promoter methylation assay

PCR-CE was performed on the Promega panel, including six mononucleotide repeat markers (BAT-25, BAT-26, NR-21, NR-24, NR-27, and MONO-27) using a Microsatellite Instability Detection Kit (Microread Gene Technology, Beijing, China). A sex locus marker (amelogenin) and one pentanucleotide repeat marker (Penta C) were analyzed as internal controls. Data analysis was performed using the GeneMapper® ID-X v.1.6 software (Applied Biosystems, Foster City, USA).

Library construction, target capture, sequencing and bioinformatics analysis were performed as previously described (*Sun et al., 2019*). Briefly, the gDNA from normal and tumor tissues was sheared into 200~300 bp fragments with a Covaris S2 ultrasonicator (Covaris, Woburn, MA, USA), and a Qseq series Bio-Fragment Analyzer (HouZe Biotech, Hangzhou, China) was used to detect the length distribution range of the products. The DNA fragments were subjected to enzymatic steps, including the NEBNext Ultra II End Repair/dA-Tailing Module (New England Biolabs, Ipswich, MA). Sequencing libraries were prepared with KAPA DNA Library Preparation Kits (Kapa Biosystems, Wilmington, MA). The libraries were hybridized into custom-designed biotinylated oligonucleotide probes (Integrated DNA Technology, Coralville, IA, USA) covering 1,021 genes (*Zhang et al., 2021*). Genomic DNA targeted sequencing was performed using a Geneplus-2000

sequencer (Geneplus-Beijing, Beijing, China). Reads were aligned to the human genome build GRCh37 using Burrow-Wheeler Aligner (BWA). Somatic single-nucleotide variant (sSNV) and indel calls were generated using MuTect (v1.14) and GATK (The Genome Analysis Toolkit, v3.4-46-gbc02625), respectively. Somatic copy number alterations were identified with CONTRA (Copy Number Targeted Resequencing Analysis, v2.0.8). Genomic rearrangements were identified by NoahCare structural variation detection (software developed in-house) to analyze chimeric read pairs. Germline genetic variants were interpreted according to American College of Medical Genetics and Genomics (ACMG) guidelines. Tumor mutational burden (TMB) was defined as the number of somatic nonsynonymous SNVs and InDels per megabase in the coding region, with a variant allele fraction of $>=0.03$. The threshold of hypermutation was calculated as previously reported by the Memorial Sloan Kettering Cancer Center (*Zehir et al., 2017*). MSI status was inferred using the MSIsensor (v0.5) software, which computes length distributions of microsatellites per site in paired normal and tumor sequence data, subsequently using these to compare the observed distribution in both samples (*Niu et al., 2014*). MSI-H was evaluated by using an empirically defined cutoff of MSI score $>8\%$, which has been validated in multiple types of cancers (*Li et al., 2022*). Raw sequence data of inconsistent cases have been deposited in NCBI SRA databases under access number PRJNA990462.

MLH1 promoter methylation testing was performed for MLH1-deficient cases. DNA was chemically modified by sodium bisulfite modification kit (Precision Scientific, Suzhou, China) to convert all unmethylated cytosines to uracils as described previously while leaving methylcytosines unaltered (*Walsh et al., 2008*). The bisulfite-modified DNA samples were then used for fluorescence real-time PCR detection. Eight known CpG dinucleotides in the MLH1 promoter were objectively identified to design primers and fluorogenic probes. Primer sequences are shown in Table S1. MLH1 methylation site was detected by human MLH1 gene methylation test kit (Precision Scientific, Suzhou, China) with FAM signal, and internal control indicated HEX signal; all amplifications included one positive control and one negative control. For the quantitative methylation-specific PCR (MSP), 5 µL of bisulfite-converted DNA was used in each amplification. PCR was performed in a reaction volume of 40 µL consisting of MLH1 buffer (including primer, probe, $Mg^{2+}$ and dNTPs) and Taq DNA polymerase at the following condition: 95 °C for 5 min, followed by 15cycles at 95 °C for 20s, 56 °C for 20s and 72 °C for 20s, then followed by 30 cycles at 95 °C for 20s, 60 °C for 30s and 72 °C for 20s. Multiplex real-time PCR was performed in ABI-7500 cycler (Applied Biosystems).

## Statistical analysis

Patient demographics of as age, family history, ascites, tumor diameter, FIGO stage, pathology subtype, grade, LVSI, muscular invasion, and lymphatic metastasis were summarized. In discordant cases, the primary reference standard was verification of NGS detection. All statistical analyses were performed using R software (version 4.2.1). Categorical variables were compared using Pearson chi-square tests to evaluate the significance of comparisons. Concordance between IHC and PCR-CE testing was evaluated by Cohen's Kappa statistics or McNemar's Chi-squared test.

# RESULTS

## Clinicopathological characteristics

A total of 333 patients with MMR/MSI status were assessed by IHC and PCR-CE for this study, and the median age of all patients was 53 years, ranging from 25 to 87 years, at the time of diagnosis of EC. Thirty of the 333 (9%) patients had a family history, including CRC, EC and other cancers. A total of 81.4% of patients were diagnosed with early-stage disease (FIGO I-II). Most patients presented with stage I (74.8%) and the endometrioid subtype (83.8%). There were 86 (25.8%) patients with LVSI. An overview of the patient clinicopathologic characteristics is summarized in Table S2.

## MMR/MSI status associations with clinicopathological features.

A total of 333 EC patients were successfully analyzed by IHC staining and PCR-CE assay in our study (Fig. 2). Of these patients, 84 (25.2%) patients presented dMMR; of these, 50 (59.5%) patients showed loss of expression of MLH1 and PMS2, 19 (22.6%) patients showed loss of expression of MSH2 and MSH6, and 7 (8.3%) and 8 (9.5%) patients showed loss of expression only of PMS2 and MSH6, respectively. For the whole tumor, 80 (24%) cases exhibited MSI-H. We investigated the relationship between MMR/MSI status and clinicopathologic characteristics and found a significant association with high-grade EC among patients whose tumors exhibited dMMR or MSI-H ($p = 0.004$ and $p = 0.003$, Table 1). Interestingly, we found that among patients <60-year-old, the proportions of dMMR *versus* pMMR tumors were 86.9% and 77.9%, respectively. Thus, dMMR patients were significantly younger than pMMR patients ($p = 0.038$). MMR status was also evaluated in both preoperative and operative specimens, but we found no correlation with the source of tumor samples ($p = 0.526$). Loss of the MSH2 or MSH6 protein was detected in numerically fewer curettage (5.8% or 8.0%, respectively) and hysterectomy (5.4% or 7.9%, respectively) specimens, whereas the MLH1 or PMS2 protein was lost in a higher fraction of the tumors (15.3% or 18.2% in curettage samples, 14.4% or 15.8% in hysterectomy samples, respectively) (Tables S3 and S4).

## Substantial agreement between MSI analysis and IHC testing in endometrial cancer

In the whole cohort, dMMR protein expression was detected in 84 (25.2%) patients, and pMMR protein expression was detected in 249 (74.8%) patients. Meanwhile, MSI was MSI-H in 80 (24.0) patients and MSS in 253 (76.0%) patients. IHC staining and PCR-CE assay results showed 100% agreement for pMMR/MSS in 249 cases and 95.2% agreement for dMMR/MSI-H (Table 2). Thus, we found four (1.2%) inconsistent cases between IHC staining and PCR-CE analysis, with a general agreement of 98.8% between the two approaches (McNemar's Chi-squared test, $p = 0.13$). In dMMR patients, we found loss of following protein lost but microsatellites in a stable state: (a) combined loss of MLH1+PMS2, (b) combined loss of MSH2+MSH6, and (c) individual loss of MSH6. By MSI analysis, half (50%) of these discrepant cases were in MSS, and the other half (50%) were MSI-L. Combined IHC and MSI analyses of 333 ECs showed an extremely high level of general agreement between the two methods (98.8%, Cohen's $\kappa = 0.98$, Fig. 3).

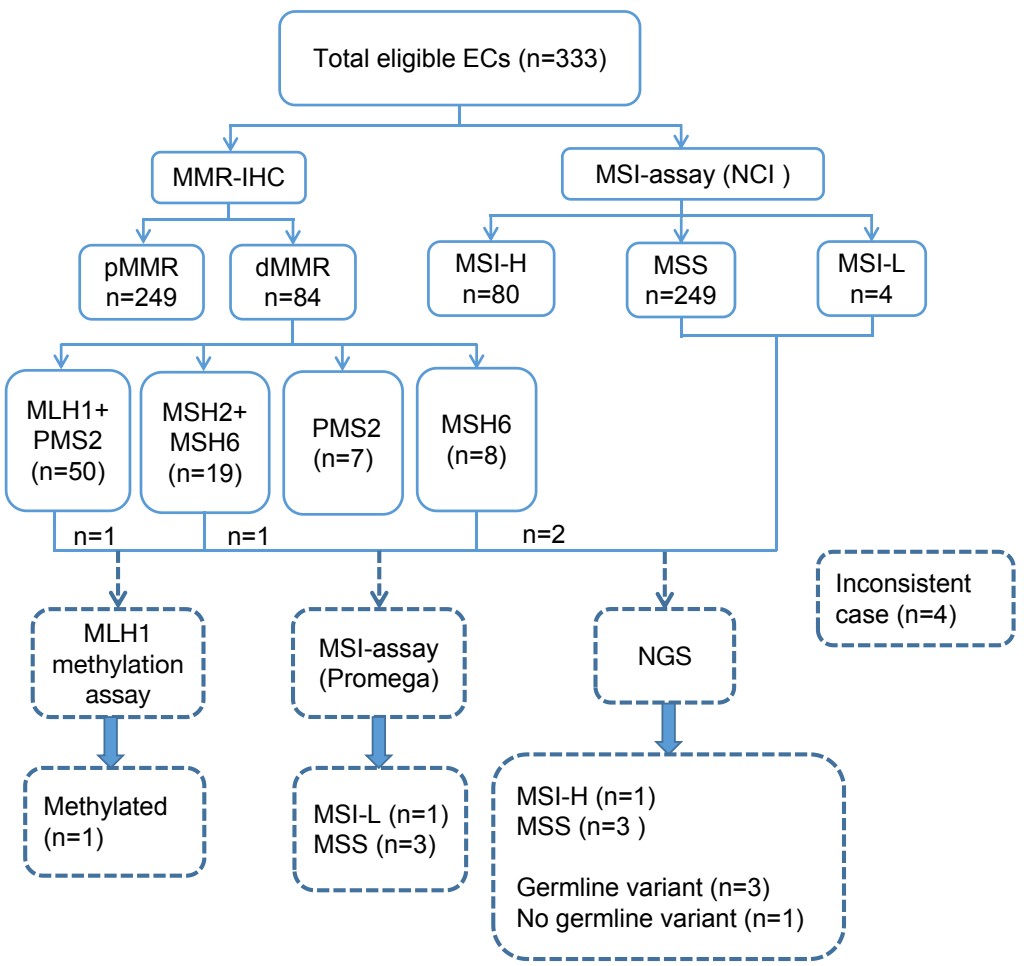

**Figure 2** **Flowchart of the study involving 333 eligible ECs.** EC, endometrial cancer; NCI, National Cancer Institute; IHC, immunohistochemistry; MMR, mismatch repair; MSI, microsatellite instability; dMMR, mismatch repair deficient; pMMR, mismatch repair proficient; MSI-H, microsatellite instability-high; MSI-L, microsatellite instability-low; MSS, microsatellite stable; NGS, next-generation sequencing.

## Analysis of discrepant cases in MMR/MSS by alternative methods

The four inconsistent cases were selected for further validation by different methods. An alternative PCR-based MSI assay (Promega panel) showed the same result in MSI analysis (NCI panel), but a different single nucleotide locus (Mono 27) was detected in one case with a loss of MSH6. As a third supplementary method to evaluate MSI status, the four discrepant cases were further tested by NGS. Based on the MSI sensor algorithm, one case result was shown to be MSI-H, and the others remained MSS (Table 3).

The molecular characteristics and clinicopathological characteristics of the four patients with dMMR/MSS are summarized in Table 4. MLH1 promoter methylation testing was performed on the first case and showed positive MLH1 promoter methylation without germline mutation and a low TMB level. Three patients had MSS tumors with lack of expression of MSH6, were further diagnosed with LS, did not meet regular MSI criteria,

**Table 1 The correlation of clinicopathological parameters with MMR protein-deficient subgroups in 333 patients with endometrial cancer.**

| | Overall N = 333(%) | dMMR N = 84(%) | pMMR N = 249(%) | p | MSI-H N = 80(%) | MSS/MSI-L N = 253(%) | p |
|---|---|---|---|---|---|---|---|
| Age | | | | 0.038 | | | 0.065 |
| <50 | 92 (27.6) | 19 (22.6) | 73 (29.3) | | 18 (22.5) | 74 (29.2) | |
| 50–59 | 175 (52.6) | 54 (64.3) | 121 (48.6) | | 51 (63.8) | 124 (49.0) | |
| >=60 | 66 (19.8) | 11 (13.1) | 55 (22.1) | | 11 (13.8) | 55 (21.7) | |
| Specimen | | | | 0.526 | | | 0.427 |
| Curettage | 131 (39.3) | 36 (42.9) | 95 (38.2) | | 35 (43.8) | 96 (37.9) | |
| Hysterectomy | 202 (60.7) | 48 (57.1) | 154 (61.8) | | 45 (56.3) | 157 (62.1) | |
| Family history | | | | 0.443 | | | 0.651 |
| No | 224 (67.3) | 57 (67.9) | 167 (67.1) | | 54 (67.5) | 170 (67.2) | |
| Yes | 30 (9.0) | 10 (11.9) | 20 (8.0) | | 9 (11.3) | 21 (8.3) | |
| FIGO stage | | | | 0.712 | | | 0.646 |
| I–II | 271 (81.4) | 70 (83.3) | 201 (80.7) | | 67 (83.8) | 204 (80.6) | |
| III–IV | 62 (18.6) | 14 (16.7) | 48 (19.3) | | 13 (16.3) | 49 (19.4) | |
| Histology | | | | 0.764 | | | 0.595 |
| EC | 279 (83.8) | 69 (82.1) | 210 (84.3) | | 65 (81.3) | 214 (84.6) | |
| Non-EC | 54 (16.2) | 15 (17.9) | 39 (15.7) | | 15 (18.8) | 39 (15.4) | |
| Grade | | | | 0.004 | | | 0.003 |
| G1/G2 | 241 (72.4) | 50 (59.5) | 191 (76.7) | | 47 (58.8) | 194 (76.7) | |
| G3 | 92 (27.6) | 34 (40.5) | 58 (23.3) | | 33 (41.2) | 59 (23.3) | |
| Lymphovascular invasion | | | | 0.419 | | | 0.590 |
| No | 247 (74.2) | 59 (70.2) | 188 (75.5) | | 57 (71.3) | 190 (75.1) | |
| Yes | 86 (25.8) | 25 (29.8) | 61 (24.5) | | 23 (28.8) | 63 (24.9) | |
| Muscular invasion | | | | 0.400 | | | 0.586 |
| Deep | 89 (26.7) | 19 (22.6) | 70 (28.1) | | 19 (23.8) | 70 (27.7) | |
| Superfical | 244 (73.3) | 65 (77.4) | 179 (71.9) | | 61 (76.3) | 183 (72.3) | |
| Lymphatic metastasis | | | | 1.000 | | | 0.951 |
| No | 297 (89.2) | 75 (89.3) | 222 (89.2) | | 72 (90.0) | 225 (88.9) | |
| Yes | 36 (10.8) | 9 (10.7) | 27 (10.8) | | 8 (10.0) | 28 (11.1) | |

**Notes.**

MMR, mismatch repair; dMMR, mismatch repair deficient; pMMR, mismatch repair proficient; MSI-H, microsatellite instability-high; MSS/MSI-L, microsatellite stable/microsatellite instability-low; FIGO, international Federation of Gynecology and Obstetrics.

and had been screened by alternative NGS assay. These patients were younger and diagnosed with EC from 49 to 57 years. Screening of case 15 was ordered because of a personal history as well as a family history of CRC.

## DISCUSSION

In this study, we comprehensively evaluated the diagnostic performance of PCR-CE (NCI panel) testing in women with endometrial cancer. Here we report a substantial agreement in MMR/MSI analysis between IHC staining and the PCR-CE assay and demonstrate that a lack of MMR protein expression is significantly associated with high-grade EC.

**Table 2  The concordance between MMR protein IHC staining and MSI analysis (NCI panel) in 333 patients of ECs.**

| | | MSI analysis | | |
|---|---|---|---|---|
| | | MSI-H | MSS/MSI-L | MSI/MMR agreement (%) |
| IHC staining | dMMR | 80 | 4 | 80/84 (95.2) |
| | MLH1/PMS2 | 49 | 1 | 49/50 (98.0) |
| | MSH2/MSH6 | 18 | 1 | 18/19 (94.7) |
| | MSH6 | 6 | 2 | 6/8 (75.0) |
| | PMS2 | 7 | 0 | 7/7 (100.0) |
| | pMMR | 0 | 249 | 249/249 (100.0) |
| | Total | \ | \ | 329/333 (98.8) |

**Notes.**

MMR, mismatch repair; dMMR, mismatch repair deficient; pMMR, mismatch repair proficient; MSI-H, microsatellite instability-high; MSS/MSI-L, microsatellite stable/microsatellite instability-low; NCI, National Cancer Institute; IHC, immunohistochemistry.

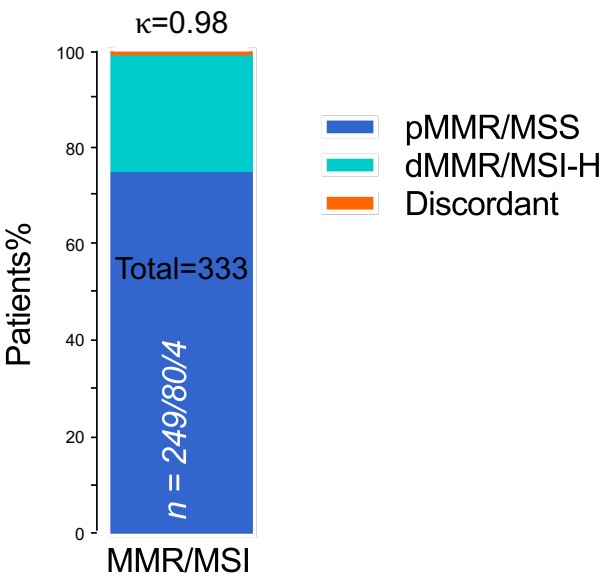

**Figure 3  Concordant MMR/MSI analysis between IHC staining and PCR testing ($n = 333$).** n = pMMR and MSS/dMMR and MSI-H/Discordant. pMMR, mismatch repair proficient; dMMR, mismatch repair deficient; MSS, microsatellite stable; MSI-H, microsatellite instability-high; Discordant, MMR protein expression was lost but MSI status was stable; $\kappa$, Cohen's kappa coefficient.

The PCR-MSI test showed 100% concordance with pMMR IHC analysis but 95.2% concordance with dMMR IHC results.

Most research has shown that 20–30% of EC patients have tumors with dMMR (*Talhouk et al., 2016*; *Kommoss et al., 2018*). Compared with pMMR tumors, patients with dMMR tumors have unique clinicopathological features, such as poor tumor differentiation, early tumor stage, and better prognosis (*Sutter et al., 2004*; *Pina et al., 2018*; *Saijo et al., 2019*). However, the correlation between MMR status and clinicopathological features

**Table 3  Analysis of the discrepant EC cases using NGS testing and PCR- MSI alternative assay.**

| Case | MMR protein expression | | | | PCR[a] | | PCR[b] | | NGS |
|---|---|---|---|---|---|---|---|---|---|
| | MLH1 | PMS2 | MSH2 | MSH6 | Status | Loci | Status | Loci | Status |
| 27 | Lost | Lost | Retain | Retain | MSS | None | MSS | None | MSS |
| 50 | Retain | Retain | Retain | Lost | MSI-L | BAT25 | MSS | None | MSS |
| 153 | Retain | Retain | Retain | Lost | MSI-L | BAT26 | MSI-L | Mono27 | MSI-H |
| 15 | Retain | Retain | Lost | Lost | MSS | None | MSS | None | MSS |

Notes.

MMR, mismatch repair; MSS, microsatellite stable; MSI-H, microsatellite instability-high; PCR, polymerase chain reaction; NGS, next generation sequencing.

[a]Based on NCI panel.

[b]Based on Promega panel.

is controversial (*Murali et al., 2019*). A systematic review has summarized the current knowledge of dMMR tumors based on IHC in EC. In total, 83% to 90% of low-grade EC tumors have the dMMR phenotype, compared with 31.4% to 77% of pMMR tumor phenotypes (*Favier et al., 2022*). Our IHC staining or PCR-CE analysis identified a correlation between MMR/MSI status and high-grade endometrioid or nonendometrioid components but not other characteristics. These disparate findings may be due to different criteria of race, stage, histological type, and testing method.

In recent years, to screen LS effectively, the National Comprehensive Cancer Network has recommended universal MMR testing for all newly diagnosed cases of EC (*Armstrong et al., 2019*). IHC staining has become the standard practice in many institutions and is a strong recommendation in new guidelines for immune checkpoint inhibitor therapy in patients with EC (*Bartley et al., 2022*). However, several other techniques, including MSI molecular analysis and NGS testing, are constantly being applied (*Favier et al., 2022*). These techniques may overcome the drawbacks of IHC staining. On the one hand, defects in the MMR system can occur secondary to genetic or epigenetic mechanisms through germline or somatic variations in one of the MMR genes or methylation of the promoter region (commonly in MLH1). On the other hand, interpreting IHC staining results is somewhat subjective. The ability to detect dMMR cases may lead to misdiagnosis by MSI testing alone, specifically MSH6 variations that tend to result in MSI-L or MSS in the tumor, which are more common in EC than CRC (*Devlin et al., 2008*; *Chui et al., 2014*). All patients in our study underwent both IHC and MSI testing, and there was a high level of consistency between the two assays. We showed that the overall deficiency rate of MMR was 25.2%, within the established range of 17–33% of all ECs (*Favier et al., 2022*), and approximately 24% of endometrial cancer patients have tumors with MSI-H by PCR-based MSI testing. Among discrepant cases, 75% of LS-dMMR cases were identified by additional testing, including MLH1 promotor methylation and NGS. The MSH6 pathogenic variation carrier rate of 100% among these cases with LS is remarkably higher than LS registry data (*Post et al., 2021*). There was no further individual screening of all dMMR patients in our study, which may have caused this bias. As up to 75% of cases do not meet the criteria of the revised Bethesda guidelines, clinical screening for personal and family history alone may miss a large proportion of women with LS (*Umar et al., 2004*). The International Society

Wang et al. (2023), *PeerJ*, DOI 10.7717/peerj.15920
**Table 4** The molecular characteristics and clinicopathological features of the discrepant cases of EC patients.

| Case | Age | FIGO | Histotype | Grade | LVSI | Personal history | Family history | Affected MMR proteins | MLH1 promotor methylation assay | TMB (Muts/Mb) | Germline variation and clinical classfication | Triaged MMRd-EC |
|------|-----|------|-----------|-------|------|------------------|----------------|----------------------|--------------------------------|---------------|----------------------------------------------|-----------------|
| 27 | 58 | III | EC | G1 | Yes | No | No | MLH1+PMS2 | Methylation | 4.80 | None | Methylated-MMRd |
| 50 | 57 | I | EC | G1 | No | No | No | MSH6 | None | 99.84 | MSH6 (NM_000179.2, Exon 4, c.1804_1805delTC); pathogenic | LS-MMRd |
| 153 | 56 | II | EC | G3 | Yes | No | No | MSH6 | None | 365.76 | MSH6 (NM_000179.2, Exon 6, c.3514dupA); pathogenic | LS-MMRd |
| 15 | 49 | I | EC | G1 | No | Yes | Yes | MSH2+MSH6 | None | 5.76 | MSH6 (NM_000179.2, Exon 4, c.651dupT); pathogenic | LS-MMRd |

**Notes.**

EC, endometrial cancer; FIGO, International Federation of Gynecology and Obstetrics; LVSI, lymphovascular invasion; MMR, mismatch repair; TMB, tumor mutation burden; MMRd, mismatch repair deficient; p53abn, p53 abnormal; LS, lynch syndrome.

of Gynecological Pathologists (ISGyP) recommends MMR/MSI status analysis (preferably with curettage or hysterectomy samples) for EC patients of all ages (*Cho et al., 2019*). Our results were similar to those of previous studies (*Talhouk et al., 2016*; *Berg et al., 2023*), with 27.5% and 26.7% of dMMR and MSI-H, respectively, for the curettage samples but slightly higher than the 23.8% and 22.3% for the hysterectomy samples.

Approximately 30% of patients with germline variations in the MSH6 gene may present with MSS (*Goodfellow et al., 2003*). However, we interestingly found that all LS-dMMR patients exhibited loss of MSH6, with MSS status in four discrepant cases. This result may be related to our insufficient sample size. Combined use of IHC and PCR-CE methods to assess MMR/MSI status can improve the accuracy of the assay (*Cho et al., 2019*; *McCarthy et al., 2019*). In the event of inconsistent results from the combined detection, a performance-proven NGS may be considered an aid for diagnosis. In previous studies, there were also a few cases of inconsistency between the two approaches, but intensive studies of these ambiguous cases have not been conducted. For a more detailed study, we examined the reasons for these discrepancies using different methods, including MLH1 methylation analysis, Promega panel MSI analysis, and NGS testing. MLH1 promoter methylation analysis revealed that the methylation to be the cause of the phenotype with dMMR but MSS status (Case 27), and this result supports the previous report (*Amemiya et al., 2022*). Case 153 was evaluated by PCR-CE with the Promega panel (Mono 27); combined with NCI panel (BAT26) results, the case was comprehensively analyzed as MSI-H status. The lower levels of MSI in the tumors of MSH6 variation carriers were related to significantly fewer unstable events with dinucleotide repeats than that with mononucleotide repeats (*Goodfellow et al., 2003*). Deep sequencing by NGS analysis revealed MSI-H status in one case (Case 153), possibly because the MSI-score was quantitatively calculated without subjectivity. Mismatch repair protein functional deficiency leads to impaired replication alignment and high-frequency base substitutions or errors, resulting in a high TMB or ultramutated state (Case 153, TMB>100 muts/Mb). However, there was a striking inverse association between MSI and TMB: MSI-H was restricted to tumors in the 10~100 mut/Mb range, while tumors with >100 mut/Mb were MSS and accompanied by replicative polymerase variations (*Campbell et al., 2017*).

IHC staining for MSH6 may frequently lead to conflicting results. Previous studies have shown that CRC patients who received neoadjuvant therapy may have an almost complete loss of MSH6 or only nucleolus staining, resulting in dMMR, but that the MSI status is MSS (*Bao et al., 2010*; *Goldstein et al., 2017*). The reasons may be secondary to subclonal mutational changes in the MSH6 gene or variations in the gene promoter. Furthermore, chemoradiation therapy may cause a decrease in cellular division and induce a resting state. Our study included one patient (Case 15) with loss of MSH6 and MSS status, with a history of CRC and neoadjuvant therapy, but NGS results showed that this was mainly due to MSH6 germline variations. Therefore, in determining MSI status, if PCR-CE results are considered false-negative, especially with isolated loss of the MSH6 protein, NGS may be helpful for ambiguous PCR-CE results. In these discrepant cases (Case 50, Case 153, and Case 15), the opportunity for immunotherapy might have been missed if the PCR-CE assay was performed using an NCI panel for EC. The College of American Pathologists and

Association Molecular Pathology (CAP and AMP) recommends that for EC patients who are considering immune checkpoint inhibitor therapy, pathologists should use MMR-IHC over MSI by PCR or NGS for dMMR testing (*Bartley et al., 2022*). Our study demonstrated a high concordance between IHC staining and PCR-CE methods for assessing MMR/MSI status in patients with EC, and the use of either detection as a screening method for MSI-H tumors is acceptable. Given the differences in the methodology of the respective assays of MMR and MSI, they are complementary in exceptional cases. For a few MSH6 germline variation cases, the tumors may not show MSI due to the functional redundancy of the MMR system, and the IHC can detect the loss of the MMR expression, thus compensating for the limitations of the PCR-CE assay.

Therefore, many challenging situations need to be recognized when reporting tumor status using MMR-IHC or PCR-CE. Previous reports indicate that the heterogeneity of the MMR/MSI status of the tumor tissue in some cases of EC might lead to ambiguous results between IHC staining and the MSI assay, which can be more visually observed with immunohistochemistry (*Saeed et al., 2021*; *Smithgall et al., 2022*; *Amemiya et al., 2022*). Minimal microsatellite marker shift has been reported in EC, less than that in MSI-H CRC, thus leading to difficulties in MMR assessment (*Wang et al., 2017*). Our study did not find a discrepant case of pMMR/MSI-H status. On the one hand, this situation may be due to an insufficient sample size. On the other hand, in the interpretation of MSI detection in EC, the microsatellite marker shift is not so evident that it leads to the determination of MSS. There are many other reasons for pMMR/MSI-H, as follows (*Watson et al., 2007*; *Laghi, Bianchi & Malesci, 2008*; *Li et al., 2013*). (a) IHC is unable to detect MMR function deficiency caused by missense mutations in the MMR gene. This often occurs in the MLH1 or MSH6 gene and will result in abnormal protein function without affecting protein translation and retaining its antigenicity. (b) Lack of detection of some other MMR protein deficiencies. (c) MSI occurring by other mechanisms, such as histone H3K36me3 recruiting hMutS $\alpha$ through direct interaction with MSH6, which regulates MMR function and manifests as MSI-H when cells lack H3K36me3 and other proteins regulating MMR function.

Assessment of MMR protein expression by IHC is cost-effective and widely available in most clinical laboratories. It can also identify affected MMR genes and effectively screen patients with suspected LS. Therefore, combined use of IHC staining and PCR-CE (NCI panel) analysis to assess MSI/MMR status may improve the accuracy of the assay. In the case of inconsistent results of the combined assay, a proven NGS method can be considered as an aid to diagnosis. Although our study involved a small sample size, we believe it is characteristic of a multiplatform detection of hospitals and provides a practical and informative dataset for most laboratories.

## CONCLUSIONS

In conclusion, we showed an overview of MMR/MSI status through IHC and PCR-CE analysis in EC of western Chinese patients and further analyzed correlations between MMR/MSI status and clinicopathological characteristics. IHC staining combined with PCR-CE analysis may effectively avoid misdiagnoses in EC patients with dMMR/MSI-H.

However, the use of PCR-CE alone to evaluate MMR/MSI status may lead to missed diagnosis, especially in EC patients with MSH6 deficiency and presenting MSS.

### Funding
The authors received no funding for this work.

### Competing Interests
The authors declare there are no competing interests.

### Author Contributions
- Cheng Wang conceived and designed the experiments, performed the experiments, authored or reviewed drafts of the article, and approved the final draft.
- Wei Kuang conceived and designed the experiments, performed the experiments, authored or reviewed drafts of the article, and approved the final draft.
- Jing Zeng conceived and designed the experiments, authored or reviewed drafts of the article, and approved the final draft.
- Yang Ren analyzed the data, prepared figures and/or tables, and approved the final draft.
- Qianqi Liu performed the experiments, prepared figures and/or tables, and approved the final draft.
- Huanxin Sun analyzed the data, prepared figures and/or tables, and approved the final draft.
- Min Feng analyzed the data, authored or reviewed drafts of the article, and approved the final draft.
- Dongni Liang analyzed the data, authored or reviewed drafts of the article, and approved the final draft.

### Human Ethics
The following information was supplied relating to ethical approvals (i.e., approving body and any reference numbers):

The study was approved by the Ethics Committee of West China Second University Hospital of Sichuan University Institutional (No.2023037)

### Data Availability
The raw data of clinical and pathologic features in 333 endometrial cancers is available in the Supplementary Files.

### Supplemental Information
Supplemental information for this article can be found online at http://dx.doi.org/10.7717/peerj.15920#supplemental-information.

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
