# Peer review of "A retrospective study of consistency between immunohistochemistry and polymerase chain reaction of microsatellite instability in endometrial cancer"

_PeerJ, doi:10.7717/peerj.15920_

## Round 0.1 · original submission · Major Revisions

The article is about the identification of mistmatch repair/microsatellite instability by immunohistochemistry and its relation to the clinical status of endometrial cancer. However, there are many issues that need to be revised and should be clearly explained by the authors.

Reviewer 1 ·

Basic reporting

In general, I think the scientific question/problem needs to be more clearly defined in the background section. The authors need to explain more clearly why this study is important. Why is it important to identify dMMR/MSI-H tumors? What are the disadvantages of the gold standard. What advantages are there of other methods? Why is it important to investigate the concordance between the gold standard and other methods? What is known about the concordance from previous studies?

I do not think the authors should go into such detail on discrepant cases in the abstract. Instead, they should focus on the general scientific question and what the study adds to the litterature.

The authors conclude MMR-IHC or PCR-based MSI analysis is proven by the study to be effective for stratifying patients for immunotherapy. However, the authors have not actually tested this. Only four out of 333 cases showed discrepancies between IHC staining and MSI assays but the authors claim they have shown that it’s necessary to re-analyze MSI status using alternative methods in these cases. This seems like a somewhat irrelevant statement due to the small number of cases. Furthermore, due to the high concordance between the methods, are both really needed? Is it really true that the combined use of these methods are proven to be effective as opposed to either method alone? Please clarify/elaborate.

“Routine MMR mutation screening for EC by immunohistochemistry (IHC) staining and/or polymerase chain reaction (PCR) MSI assay to enable selection of immune checkpoint inhibitor therapy, screening for Lynch syndrome, and prognostic guidance for molecular classification of endometrial cancer”- The grammar of this sentence is incorrect- please clarify.

Methods
- “Tumor tissues with surrounding normal tissues were stained with nuclei, and no background nonspecific staining was interpreted as retained staining”- unclear grammar, please clarify.
- “((H&E)-stained” should be (H&E)-stained. The extra bracket should be removed.

Results
- “Information about family history was found”- I think the authors mean to say that these patients had a family history of endometrial cancer. The sentence needs to be clarified.
- “An overview of the patient clinicopathologic characteristics was summarized in Table S1”- better to use present tense.
- “Interestingly, within dMMR tumors, we found that patients (<60 years old, 86.9%) were younger
than the pMMR subgroup (<60 years old, 77.9%; p=0.038)”- Less confusing to write “86.9% were <60 years old” etc.
- Loss of MSH2 and MSH6 was detected in fewer curettage and hysterectomy specimens. Statistically fewer (ie significant p-value?) or just numerically fewer?

Discussion
- In general, the grammar breaks down in the discussion. Here, the language needs a major overhaul. Please revise the grammar carefully.
- There was a high general agreement between methods. Can the authors comment on the implications of this as this seems to have been the main scientific question of the study.
- Do not start sentences with numbers.
- “Our results showed that curettage and hysterectomy samples had a similar conclusion in MMR and
- MSI status evaluating, presenting the MMR deficiency rate of 27.5% and 23.8%, but slightly higher than the MSI-H rate of 26.7% and 22.3%”- unclear grammar. Please revise.
- “However, our interesting finding was that in the four discrepant cases, all LS-dMMR patients exhibited loss of MSH6 but MSS status.”- However, we interestingly found that..”
- “Another method was used to determine which is accurate in contradictory results (these were judged as dMMR by IHC staining but MSS by PCR-based MSI”- unclear grammar

Experimental design

No further comments

Validity of the findings

No further comments

·

Basic reporting

The quality of English language is relatively satisfactory, while there are several unclear, incomprehensible, or ambiguous phrases and sentence:
- sentence "germline mutations in the significant genes encoding the four proteins of the MMR system lead to loss of protein expression" is imprecise since it is unknown how, for instance, a germline missense variant can cause "loss of protein expression"
- phrase "MMR status deficiency" is meaningless, proper is "MMR deficiency status", "MMR status" or "MMR deficiency"
- line 109: proper is "retained or loss", expression cannot be both retained and loss
- line 125-126: sentence should be re-written in past participle tense, not like a protocol
- line 208-209: proper would be "detected in one case with a loss of MSH6"
- line 218: proper would be "Screening of case 15 had been ordered because of..."
- line 249: it is unclear what means "path_MSH6 carrier"
References are appropriate and sufficient.
There should be more information about MMR process in 'Introduction'.
Article is quite professionally structured, figures and tables are appropriate, but all abbreviations presented in figures and tables must be explained in figure legends (that is missing for Figures 1 and 2) and tables' footnotes. Also, for Figure 3 a sample size (n) should be put in legend, now it is non-intuitive.
Per-sample results were provided, but NGS raw data has not been.
All hypotheses were corroborated by obtained results.

Experimental design

Proper number of adequate methods were applied, and research questions were well defined.
Unfortunately, the biggest drawback of the study behind this manuscript is its irreproducibility because lots of important details are missing from methods' descriptions:
- for all methods the exact model of used instrument must be precisely stated, as well as exact name of all used commercial kits
- NGS must be re-written from scratch with ALL details so anyone can repeated it, this is especially important for NGS data analyses, since it is unclear what is "MSI sensor algorithm", how were variants classified as pathogenic/functional, etc.
- it is unclear how results of "MLH1 promoter methylation" were quantified or qualified, i.e., is even one un/methylated CpG island significant

Validity of the findings

Majority of requirements regarding the "Validity of the findings" have been achieved, however statement regarding the p53 abnormal staining came out of nowhere at the end of 'Results'. This must be put in proper context, e.g., 'Introduction', 'Methods', etc.
Line 242: "the drawbacks of IHC staining" should be discussed in more detail.
Line 277-278: It is unclear how neoadjuvant chemotherapy may result in the absence of MSH6 expression?!
It is unclear if statistical analyses on "Family history" (Table 1) were performed taking into account only "No" and "Yes" categories, since taking into consideration also "Unknown" is meaningless.

Additional comments

All abbreviations should be explained after first mentioning (e.g., NCI, pMMR, FIGO, LS, ISGyP, TMB, etc.)
Word "mutation" should generally be avoided and rather used "variation", "pathogenic variation", "benign variation", "missense variation", etc.
Phrase "PCR-MSI" is misleading and in 'Methods' it should be mentioned that it is in fact PCR followed by fragmental analysis (capillary electrophoresis).
Line 176: Tumors with TMB>100 are so called ultramutated, while those with TMB>10 are hypermutated (DOI:10.1016/j.cell.2017.09.048).

---

## Round 0.2 · accepted · Accept

The authors have satisfactorily responded to all the concerns and questions of the reviewers and appropriately improved the quality of this article.

Reviewer 1 ·

Basic reporting

No further comments

Experimental design

No further comments

Validity of the findings

No further comments

Additional comments

The authors have satisfactorily addressed the comments.

·

Basic reporting

Authors have satisfactorily responded to all my concerns and questions, and appropriately improved quality of this manuscript.

Experimental design

Authors have satisfactorily responded to all my concerns and questions, and appropriately improved quality of this manuscript.

Validity of the findings

Authors have satisfactorily responded to all my concerns and questions, and appropriately improved quality of this manuscript.

Additional comments

Authors have satisfactorily responded to all my concerns and questions, and appropriately improved quality of this manuscript.